# A Retrospective Time and Value Analysis of Surgical Oncology Cases Using 3D Printing: A Comprehensive Cancer Center Experience

**DOI:** 10.3390/bioengineering12080821

**Published:** 2025-07-30

**Authors:** Sujaya H. Rao, James Harris, Lumarie Santiago, Paige D. Brown, Justin Bird, Karthik Tappa

**Affiliations:** 1Division of Diagnostic Imaging, The University of Texas M. D. Anderson Cancer Center, Houston, TX 77030, USA; srao@mdanderson.org (S.H.R.); jmharris@mdanderson.org (J.H.); 2Department of Breast Imaging, The University of Texas M. D. Anderson Cancer Center, Houston, TX 77030, USA; lumarie.santiago@mdanderson.org; 3College of Pharmacy and Health Sciences, Campbell University, Buies Creek, NC 27546, USA; brownp@campbell.edu; 4Department of Orthopedic Oncology, The University of Texas M. D. Anderson Cancer Center, Houston, TX 77030, USA; jebird@mdanderson.org

**Keywords:** 3D printing, surgical oncology, procedure time, value analysis

## Abstract

**Introduction:** The use of 3D-printed models in surgical planning has gained traction in light of its potential to improve precision and patient outcomes. The objective of this study was to review data and provide a time and value analysis of the use of 3D printing at a National Cancer Institute (NCI)-designated comprehensive cancer center. The estimated time of surgical procedures for surgical planning was compared with the time required for procedures that did not use 3D printing. Providers who used 3D printing completed surveys, and then the results of said surveys were analyzed to assess the value of 3D printing. **Materials and Methods:** Electronic health records were reviewed for patients who underwent hemipelvectomies with and without 3D printing. A list of 20 observations involving 3D printing was used as a baseline sample and matched with another 20 observations that did not utilize 3D printing. Electronic health records were reviewed to obtain mean estimates of the procedure time. The data was collected and analyzed between January 2018 and April 2025. **Results:** The mean surgery time for procedures using 3D printing was 868 min, compared to 993 min for procedures that did not utilize 3D printing. In contrast, the median procedure times were 907.5 min for procedures using 3D printing and 945.0 min for those that did not utilize 3D printing. Most providers (85.7%) felt that using 3D-printed models or guides was important. Similarly, 80% responded that using a 3D-printed model or guide saved them time, and another 73.3% responded that after using the 3D-printed model, they were confident in their treatment plan. **Conclusions:** Using 3D printing for surgical cases at the comprehensive cancer center saved procedure time and added value for the surgeons.

## 1. Introduction

Surgical oncology is the field of cancer care that focuses on the surgical treatment and management of cancer. Diagnostic imaging plays a pivotal role in surgical oncology, serving as the foundation for accurate diagnosis, preoperative planning, and postoperative evaluation. Diagnostic imaging is critical for surgeons, allowing them to delineate the extent of disease [1,2]. The presurgical planning conducted by the surgeon relies on a patient’s medical imaging, such as computed tomography (CT) or magnetic resonance imaging (MRI). Surgical oncologists review the patient’s imaging results before surgery to identify the area targeted for removal [3]. CT or MRI are the conventional imaging modalities used before surgery. These imaging modalities aid in determining resection ability, guiding minimally invasive approaches, and mapping critical anatomical structures. Advanced imaging techniques, including diffusion-weighted MRI and PET-CT, offer functional insights into tumor metabolism and vascularity, aiding in the differentiation between malignant and benign tumors [4]. By providing real-time, detailed anatomical and functional data, diagnostic imaging not only enhances the surgeon’s ability to plan effective oncological procedures but also plays a crucial role in evaluating treatment responses and detecting recurrence, ultimately improving patient outcomes in surgical oncology [5]. Although CT and MR images provide information about the spatial relationships between the disease and a patient’s normal anatomy, the display format of this data can hinder the appreciation of these relationships as they exist within the patient. The visualization of a patient’s anatomy on a flat-screen computer monitor limits the comprehension of complex anatomy and structures [6]. As the intent of surgery is curative, and the aim is to completely remove the cancer, understanding the extent of disease and the relationships between surrounding structures depicted by CT and MRI is critical. Translating this information, which is traditionally displayed in a two-dimensional format, to a format akin to the surgeon’s three-dimensional intraoperative experience may facilitate better understanding [6].

Three-dimensional printing is a manufacturing method in which objects are made by fusing materials such as plastic, powders, ceramics, metal, or even living cells [7]. The process includes creating a digital model using dedicated software to outline diseased and normal anatomy. The resulting computer file contains the instructions that will allow the 3D printer to make the digital model into a physical model. Specific to surgical oncology, patient imaging results such as CT or MRI scans are used as the source of data to produce a physical three-dimensional model of the cancer surgical site to assist the surgeon in preparing for surgery. Computer-aided design (CAD) software can also be used to help the surgeon designate or prescribe a surgical plan and visualize it onto the anatomy already outlined, thus providing an opportunity for rehearsal. These hand-held models (shown in Figure 1) can aid the surgical oncologist in visualizing the patient’s anatomy, guiding their surgical approach, and helping to anticipate problems that may arise in the operating room. Research by Pugliese et al. established that touching the physical 3D-printed model, or tactile response, significantly improved the surgeon’s understanding of structures and spatial orientation [8].

The adoption of 3D-printing technology in healthcare has rapidly expanded over the past decade, offering a transformative approach to surgical planning, prosthetic development, and patient education. This technology has shown tremendous potential in facilitating preoperative planning for intricate procedures, where a precise understanding of anatomical structures is critical for success. Despite its growing use, there remains a need to evaluate its economic and operational impact on point-of-care facilities [9].

Operating room (OR) time is one of the hospital’s most significant cost drivers and predictors of health outcomes. Any intervention that streamlines surgical workflows or reduces procedural time has the potential to result in substantial financial savings while maintaining or improving patient outcomes [10,11].

There is variability in the OR cost per minute per hospital, which ranges from USD 37 to USD 100 per minute. However, this expense can represent a considerable opportunity cost, resulting in lost revenue due to inefficiency [9]. A meta-analysis examining new technology—specifically the use of 3D printing–assisted surgery for distal radius fractures—found that the overall procedure time was significantly reduced (157 min versus 165 min) compared to conventional methods [12]. Prolonged anesthesia duration with longer procedure times is also associated with increased odds of complications [13]. The integration of 3D printing presents both opportunities and challenges. Institutional stakeholders require clear evidence of downstream impacts on operating room efficiency, resection margin status, complication rates, length of stay, and overall cost of care. While upfront expenditure related to software, hardware, and labor can be quantified, their relative value must be assessed in the context of broader clinical and economic outcomes.

The overall purpose of this study was to examine the procedure time and value analysis of using 3D-printed models at a comprehensive cancer center. This retrospective, mixed-methods research study evaluated quantitative and qualitative data gathered upon completion of surgical cases with and without the use of 3D-printed models. By comparing cases with and without 3D printing for hemipelvectomy procedures, the aim was to provide a holistic assessment of how 3D printing contributes to high-value cancer care.

## 2. Methods

### 2.1. Study Design

The total procedure time of the surgical procedures provided quantitative data for the time analysis. Feedback from a survey administered to the surgeon who requested the 3D-printed models provided both quantitative and qualitative data for value analysis.

The specific aims of this study were as follows:To quantitatively analyze the total procedure time required for hemipelvectomy procedures that used 3D printing for surgical planning.To quantitatively analyze the total procedure time required for hemipelvectomy procedures that did not use 3D printing for surgical planning.To qualitatively and quantitatively assess the value of 3D printing by analyzing the surveys completed by the providers who utilized 3D printing.

### 2.2. Study Design Rationale

The increasing adoption of 3D-printed models in hospitals presents a significant opportunity to enhance clinical workflows and patient outcomes. The study designs of the current literature in this field range from observational studies to prospective randomized trials. However, previous trials did not focus on procedure times or the value of 3D printing performed at a comprehensive cancer center. To address this gap, this study was designed to evaluate the initial sample size currently available and to take a deeper dive into procedure time and its value related to 3D printing. This study adopted a retrospective mixed-methods design, combining quantitative and qualitative approaches, to capture both measurable procedure duration data and subjective value perceptions.

### 2.3. Population

The population for this study included patients who were diagnosed with cancer and underwent hemipelvectomy as part of their treatment plan. By evaluating cases with and without the use of 3D-printed models, this study isolated the specific impacts of this technology on key metrics, including procedure time. To provide a holistic view, the study included stakeholder surveys. This qualitative data helped contextualize insights into perceived value, as well as the practical and clinical utilities of the 3D-printed models.

### 2.4. Sampling Plan

The study population consisted of adult patients (18 years of age or older) who had undergone hemipelvectomies at the comprehensive cancer center between January 2018 and April 2025, with and without the use of 3D-printed models. A total of 40 procedures were included in this study. Pregnant women and cognitively impaired adults were also included if 3D printing was used for their cancer surgery.

Inclusion Criteria:(1)18 years of age or older.(2)Underwent surgery between January 2018 and April 2025.(3)Underwent radical resection of tumor on the pelvis and hip joint (CPT code 27076).

EPIC is the electronic health record system at the comprehensive cancer center that we used for this study. Electronic health records were reviewed for patients who underwent surgical treatments with and without 3D printing. Cases that used 3D printing were filtered based on CPT codes for 3D printing, namely 0559T, 0560T, and procedure CPT code 27076, between January 2018 and April 2025. The data collected included the name of the procedure and the length of the procedure. No demographic or personal health information was collected.

A list of 20 hemipelvectomies involving 3D printing was filtered from a list of 56 which met eligibility criteria and was used as a baseline sample. Another 20 similar procedures that did not utilize 3D printing were collected as comparison data. Appendix A, Table A1, shows the time taken in minutes for cases performed using with and without 3D printing. This matching set was identified by filtering cases that applied to CPT code 27076. Further analysis was conducted to ensure that both cohorts were comparable.

Information about the value of 3D printing as an adjunct to surgical procedures was qualitatively gathered and analyzed with descriptive statistics from the data collected from an approved survey. Due to the retrospective nature of this study, a 15% response rate from the participants was anticipated. A 15% response rate was acceptable as only descriptive statistics were analyzed.

### 2.5. Measurement/Instruments

An assessment survey was conducted using REDCap (Research Electronic Data Capture) in accordance with routine operations of the 3D printing program. REDCap is a tool hosted at the comprehensive cancer center described in this study and is a secure, web-based application with controlled access designed to support data capture for research studies. Every provider who ordered a 3D-printed model or guide received a survey via REDCap. The survey included items assessing the ease/difficulty of using a 3D-printed model or guide, its impact on the level of confidence or refinement of the treatment plan, its importance, and the time savings it provided. One item assessing radiation exposure and one open-ended question about any changes to the 3D-printed model/guide were also included. The survey had built-in logic interactions, drop-down choices, (Appendix B) and was automatically administered to every provider after the 3D-printed model was delivered. The survey included 14 questions with a 5-point Likert scale (strongly disagree, disagree, neither agree nor disagree, agree, or strongly agree) to assess the value of 3D-printed models for surgical cases. A 5-point Likert scale is a reliable and valid instrument for measuring attitudes or opinions and is often used in social sciences research [14].

## 3. Results

Given that this is a retrospective, exploratory study involving data limited only to medically necessary procedures performed at our institution, certain statistical assumptions could not reliably be met. Accordingly, we analyzed our data with a nonparametric Mann–Whitney test and chose a wider confidence interval of 90% [15]. Table 1 shows that there was a statistically significant difference in the median procedure duration between those that did and did not utilize 3D printing (*p* < 0.05). Assisted procedures which incorporated 3D printing had a median duration of 907.5 min compared to 945.0 min for those that did not utilize this technology.

While inferential means testing could not be performed, the descriptive statistics are still noteworthy. Table 2 shows a difference of 125 min, or 12.6%, between the mean procedure time for 3D printing-assisted procedures and that of those that did not utilize 3D printing. We hope our research can inform future studies with a larger sample that can utilize inferential means testing.

Surveys were also sent to surgeons who have utilized 3D-printing services in the past two years. A 15% response rate was anticipated due to the retrospective nature of the study. A total of 14 complete surveys, or a 14% response rate, were recorded. There was a larger pool of potential respondents than in inferential analysis because comparison data were not required for qualitative analysis. Figure 2, Figure 3 and Figure 4 show the results of this survey. Most providers (85.7%) felt that the use of 3D-printed models or guides was important to their case. Similarly, 80% responded that the use of a 3D-printed model or guide saved them time, and another 73.3% responded that after using the 3D-printed model, they were confident in their treatment plan.

## 4. Discussion

Utilizing 3D printing to assist with surgical planning and other medical procedures is a relatively new phenomenon. The 3D printing lab at our institution has been active since 2021. In this time, a total of 56 cases have used this new technology. Of the 56 procedures performed at our institution, we were able to select 20 that met our criteria for analysis and comparison. Recent scientific literature and data management frameworks highlight that small sample sizes compromise the ability to accurately extrapolate findings to broader populations, as estimates of statistical relationships become highly unstable with a reduced amount of data. This limitation must be carefully considered when interpreting the results of this study [16]. However, other researchers have demonstrated that smaller datasets with domain expertise and meaningful selection can yield superior results to larger data, so bigger may not always be better [17]. While a means test would have been ideal, we were unable to confirm a Gaussian distribution and thus chose the nonparametric Mann–Whitney U test for analysis [15]. We deemed a 90% confidence interval as acceptable because the study is retrospective, exploratory, and seeks primarily to inform future research. As such, a lower confidence interval does not increase patient risk.

This study retrospectively reviewed the procedure time of hemipelvectomies, comparing two cohorts of patients. A total of 56 cases were analyzed, of which 20 procedures utilized a 3D-printed model to aid in surgical planning, and 20 procedures were performed without the assistance of a 3D-printed model. Survey results collected to assess the value of 3D printing for surgical cases were also analyzed. Ryan et al. conducted a similar retrospective study analyzing the length of time spent in the operating room using cardiac 3D-printed models. They compared eight standard-of-care cases with two which used 3D-printed anatomic models. The mean time spent in the operating room and the case length were shorter when the treatment was planned using an anatomical model dealing with highly complex disease lesions with a mean treatment time greater than 90 min. Unlike in our study, the length of time was not found to be statistically significant [18].

Previous publications have found that increased time under anesthesia can lead to increased risk of complications [13]. The results from this study noted statistically significant reductions in average procedure time when using 3D-printed models. This study included a limited sample size, since the 3D printing program at this institution launched in late 2021.

The findings of this pilot study pave the way for future research into the broader applications of 3D printing across various medical disciplines. Future studies could include various procedure types and study them longitudinally. Additionally, examining the impact of 3D printing on patient outcomes over the long term, including metrics such as complication rates, readmissions, and recovery times, could provide deeper insights into its value.

## Figures and Tables

**Figure 1 bioengineering-12-00821-f001:**
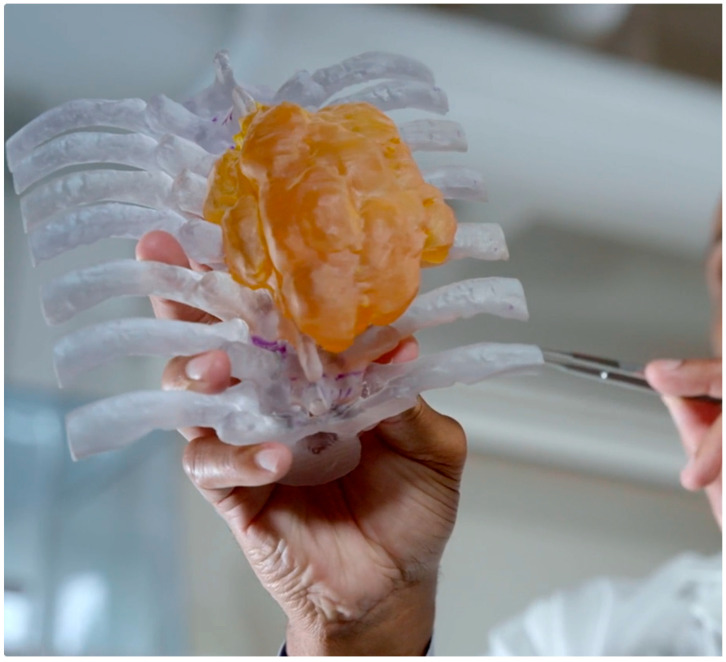
Hand-held 3D-printed model showing tumor on spine.

**Figure 2 bioengineering-12-00821-f002:**
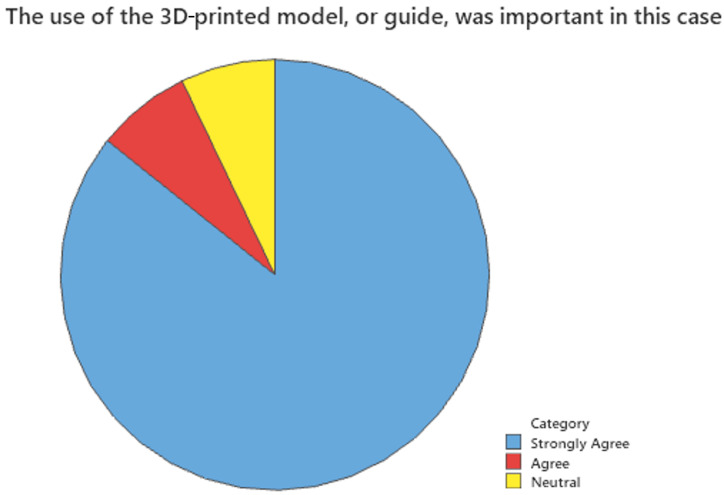
The use of the 3D-printed model or guide was important in this case.

**Figure 3 bioengineering-12-00821-f003:**
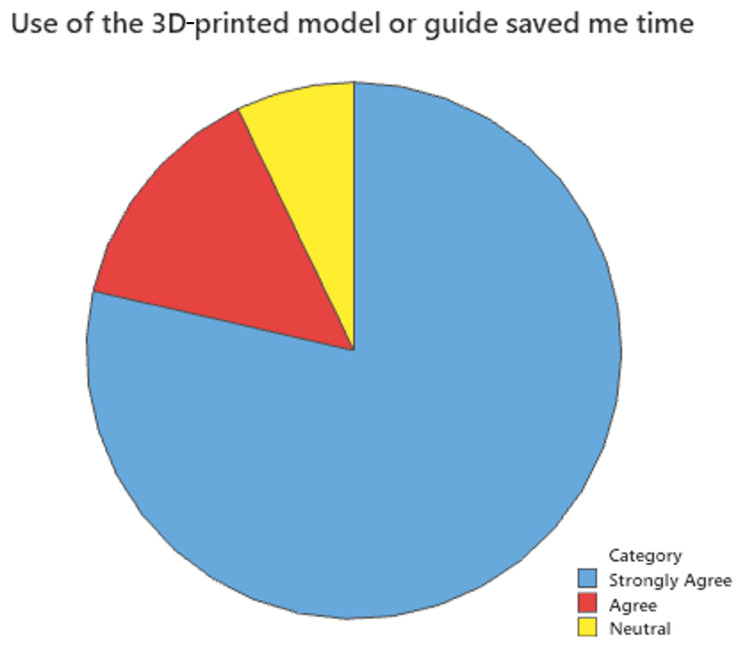
Use of the 3D-printed model or guide saved me time.

**Figure 4 bioengineering-12-00821-f004:**
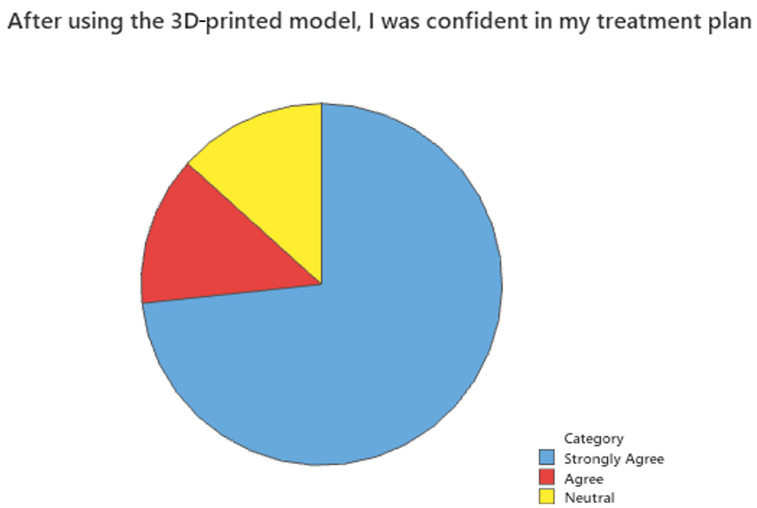
After using the 3D-printed model, I was confident in the treatment plan.

**Table 1 bioengineering-12-00821-t001:** Achieved confidence interval and Mann–Whitney test results.

Estimation for Difference			Test		
			Null hypothesis		*H*_0_: η_1_ − η_2_ = 0
			Alternative hypothesis		*H*_1_: η_1_ − η_2_ ≠ 0
**Difference**	**CI for** **Difference**	**Achieved** **Confidence**	**W-Value**	* **p** * **-Value**	
−91	(−209, −17)	90.38%	333.00	0.039	

**Table 2 bioengineering-12-00821-t002:** Descriptive mean data.

Descriptive Statistics				
**Sample**	**N**	**Mean**	**StDev**	**SE Mean**
3D Duration	20	868	133	30
Non-3D Duration	20	993	146	33

## Data Availability

The raw data supporting the conclusions of this article will be made available by the authors on request.

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
