# Peer review of "A Retrospective Time and Value Analysis of Surgical Oncology Cases Using 3D Printing: A Comprehensive Cancer Center Experience"

_bioengineering, 2025, doi:10.3390/bioengineering12080821_

Round 1

Reviewer 1 Report

Comments and Suggestions for Authors

The paper has a potential but also various deficiencies which succinctly are:

  • The paper is too brief, it should be given more space both to the Introduction and to the Discussion
  • The abstract is written based on the structure of a medical journal, it should be changed
  • There is no methodological comparison with other references
  • From a  statistical viewpoint there are very few data. Can Authors confirm that their distribution is Gaussian or what?
  • The final results are not clear. Are they objective or subjective?
  • I did not find confidence interval
  • The subjective evaluation (Likert scale) again is partial and not well developed
  • At least, a discussion on the subjectivity and limitations of employed data would be necessary. To cite the following:
    • AAVV, Is bigger always better? A controversial journey to the center of machine learning design, with uses and misuses of big data for predicting water meter failures, J Big Data, 2019, doi: 10.1186/s40537-019-0235-y

    • Laney, D. (2001). 3D Data Management: Controlling Data Volume, Velocity and Variety. META Group Research Note

Author Response

Thank you so much for taking the time to review our manuscript, and for your thoughtful feedback.

Reviewer #1          

Comment 1: The paper is too brief; it should be given more space both to the Introduction and to the Discussion.

Response 1: We appreciate your comment suggesting providing in-depth research on this topic. We have added the paragraphs below in the introduction section of the manuscript giving more background to 3D printing in point-of-care facilities.

Surgical oncology is the field of cancer care that focuses on the surgical treatment and management of cancer. Diagnostic imaging plays a pivotal role in surgical oncology, serving as the foundation for accurate diagnosis, preoperative planning, and postoperative evaluation. Diagnostic imaging is critical to surgeons to delineate the extent of disease.1,2 Presurgical planning conducted by the surgeon relies on a patient’s medical imaging such as computed tomography (CT) or magnetic resonance imaging (MRI). Surgical Oncologists use the patient’s imaging before surgery to identify the area targeted for removal. 3  CT or MRI are the conventional imaging modalities used before surgery. These imaging modalities aid in determining resection ability, guiding minimally invasive approaches, and mapping critical anatomical structures. Advanced imaging techniques, including diffusion-weighted MRI and PET-CT, offer functional insights into tumor metabolism and vascularity, aiding in the differentiation between malignant and benign tumors.4 By providing real-time, detailed anatomical and functional data, diagnostic imaging not only enhances the surgeon’s ability to plan effective oncological procedures but also plays a crucial role in evaluating treatment responses. 5 and detecting recurrence, ultimately improving patient outcomes in surgical oncology. Although CT and MR images provide information about the spatial relationships between the disease and the normal anatomy, the display format of this data can hinder the appreciation of these relationships as they exist within the patient. The visualization of a patient’s anatomy on a computer flat screen monitor limits the comprehension of complex anatomy and structures. 6 As the intent of surgery is curative by completely removing the cancer, understanding the extent of disease and relationships between surrounding structures depicted by CT and MRI is critical. Translating this information traditionally displayed in a 2-dimensional format to a format akin to the 3-dimensional intraoperative surgeon experience may facilitate better understanding. 6

Additionally, In the Discussion section, we have included below paragraphs discussing in-depth various analyses performed in this study, and reasons for choosing these types of analysis for the study.

“Utilizing 3D printing to assist in surgical planning and other medical procedures is a relatively new phenomenon. The 3D printing lab at our institution has been active since 2021.  In this time a total of 56 cases have used this new technology. Of the 56 procedures performed at our institution, we were able to select 20 that met our criteria for analysis and comparison. Recent scientific literature and data management frameworks highlight that small sample sizes compromise the ability to accurately extrapolate findings to broader populations, as estimates of statistical relationships become highly unstable with reduced data This limitation must be carefully considered when interpreting the results of this study. 16 However, other researchers have demonstrated that smaller datasets with domain expertise and meaningful selection can yield superior results to larger data, so bigger may not always be better. 17 While a means test would have been ideal, we were unable to confirm a gaussian distribution and thus chose the nonparametric Mann-Whitney U test for analysis. 15 We deemed a 90% confidence interval as acceptable because the study is retrospective, exploratory, and seeks primarily to inform future research. As such, a lower confidence interval does not increase patient risk.

This study retrospectively reviewed the procedure time of hemipelvectomies, comparing two cohorts of patients. A total of 56 cases were analyzed, of which 20 procedures utilized a 3D-printed model to aid in surgical planning, and 20 procedures were performed without 3D-printed model assistance. Survey results collected to assess the value of 3D printing for surgical cases were also analyzed. Ryan et al conducted a similar retrospective study analyzing the length of time in the operating room using cardiac 3D printed models. They compared 8 standard of care cases with 2 which used 3D printed anatomic models. The mean time for the operating room and case length were less when the case was planned with an anatomical model with highly complex disease lesions with mean time greater than 90 min. Unlike our study, the length of time was not found to be statistically significant. 18  

Comment 2: The abstract is written based on the structure of a medical journal, it should be changed

Response: As this is a retrospective analysis study, we have strictly adhered to the guidelines of MDPI’s bioengineering journal template and used similar structure to organize our abstract.

Comment 3: There is no methodological comparison with other references

Response: We agree with you. We now have added below methodological comparison paragraphs with respect to another similar study in this field.

“Ryan et al conducted a similar retrospective study analyzing the length of time in the operating room using cardiac 3D printed models. They compared 8 standards of care cases with 2 which used 3D printed anatomic models. The mean time for the operating room and case length were less when the case was planned with an anatomical model with highly complex disease lesions with mean time greater than 90 min. Unlike our study, the length of time was not found to be statistically significant.18

 Previous publications have found that increased time under anesthesia can lead to increased risk of complications 26. The results from this study noted statistically significant reductions in average procedure time when using 3D-printed models. This study included a limited sample size since the 3D printing program at this institution launched in late 2021.”

Comment 4: From a statistical viewpoint there are very few data. Can Authors confirm that their distribution is Gaussian or what?

Response 4: We commend the reviewer for suggesting the appropriate statistical tools for the analysis of this study and requesting more details about the analysis performed in the manuscript. We have instead performed the non-parametric Mann Whitney analysis for the data and added below in the results section.

“Given that this is a retrospective, exploratory study with data limited only to medically necessary procedures performed at our institution, certain statistical assumptions could not reliably be met. Accordingly, we analyzed our data with a nonparametric Mann-Whitney test and chose a wider confidence interval of 90%.15 Table 1 shows that there was a statistically significant difference in median procedure duration between those that did and did not utilize 3D Printing (p<0.05). 3D printed assisted procedures had a median duration of 907.5 minutes compared to 945.0 minutes for those that did not.”

While inferential means testing could not be performed, the descriptive statistics are still noteworthy. Table 2 shows a difference of 125 minutes, or 12.6%, between mean procedure time for 3D Printing assisted procedures and those that did not utilize 3D printing We hope our research can inform future studies with a larger sample that can utilize inferential means testing.

Comment 5: The final results are not clear. Are they objective or subjective?

Response 5: The results are mixed.  Survey responses are inherently subjective. For this study, the data were retrospectively analyzed from an ongoing survey conducted through the American College of Radiology (ACR) 3D Printing Registry. Here is the link to the survey. ACR 3D Printing Registry  This survey is standardized and is currently being used nationwide.

The Mann-Whitney test results are objective.  We were comfortable using the less powerful 90% confidence interval as this is an exploratory, retrospective study that offers opportunities to validate with future prospective studies.

Comment 6: I did not find confidence interval

Response 6: Thank you so much for providing us with such a constructive suggestion. We have added the paragraph below in the results section stating the deemed confidence level used in the study. We deemed a 90% confidence interval acceptable because the study is retrospective, exploratory, and seeks primarily to inform future research. As such, a lower confidence interval does not increase patient risk.

Comment 7: The subjective evaluation (Likert scale) again is partial and not well developed

Response 7: Thank you so much for this comment. While it's true that the Likert scale can be subjective, the Likert scale was selected by the American College of Radiology (ACR) in their survey tool for sites (such as the one being analyzed in this study) participating in the national 3D printing registry (ACR 3D Printing Registry).  The data Likert scale is widely used due to its ease of use, and the ability to convert subjective data into quantifiable metrics.

Comment 8: At least, a discussion on the subjectivity and limitations of employed data would be necessary. To cite the following:

    • AAVV, Is bigger always better? A controversial journey to the center of machine learning design, with uses and misuses of big data for predicting water meter failures, J Big Data, 2019, doi: 10.1186/s40537-019-0235-y
    • Laney, D. (2001). 3D Data Management: Controlling Data Volume, Velocity and Variety. META Group Research Note

Response 8: Thank you for your comment and suggesting these great references. We have included both citations (#16 and #17) as drafted below within the discussion section to help justify our small sample size: Below paragraph with suggested references is now in the Discussion section of the manuscript.

“Recent scientific literature and data management frameworks highlight that small sample sizes compromise the ability to accurately extrapolate findings to broader populations, as estimates of statistical relationships become highly unstable with reduced data. This limitation must be carefully considered when interpreting the results of this study.16   However, other researchers have demonstrated that smaller datasets with domain expertise and meaningful selection can yield superior to results to larger data, so bigger may not always be better.17

Reviewer 2 Report

Comments and Suggestions for Authors

The submitted manuscript does not contain a research section. Here is a report on the successes of the center. Perhaps the text should be titled not "Article", but "Report" or "Concept".
Yes, the presented information may be interesting from a popular point of view on the effectiveness of using 3D printing, but what clinical cases, what the methodology is, this is all beyond the scope of the manuscript.
I leave the decision on acceptance to the discretion of the academic editor.

Author Response

Dear reviewer, thank you so much for taking time and giving us valuable feedback. Please see below our response to your comments.

Reviewer #2:

The submitted manuscript does not contain a research section. Here is a report on the successes of the center. Perhaps the text should be titled not "Article", but "Report" or "Concept".
Yes, the presented information may be interesting from a popular point of view on the effectiveness of using 3D printing, but what clinical cases, what the methodology is, this is all beyond the scope of the manuscript.

I leave the decision on acceptance to the discretion of the academic editor.

Response:

Thank you very much for your thoughtful and constructive feedback.
We would like to clarify that our manuscript was designed as a retrospective time and value analysis of 3D printing integration in surgical oncology, focusing on impact of 3D printing in surgery and institutional experience rather than specific clinical outcomes.

However, if the editorial team believes this work is more appropriate as a clinical report or concept implementation, we are open to reclassifying the submission accordingly.

In terms of methodology, the manuscript included data from hemipelvectomy procedures over time, focusing on effectiveness of utilization of 3D printing by comparing operation room time of patients undergoing resection procedures with and without 3D printing services. We respectfully would like to note that the current manuscript was intentionally designed as a retrospective time and value analysis, rather than a clinical case series or interventional trail. The primary focus is on procedural time, and institutional process improvement in the integration of 3D printing in oncology workflows.

This approach aligns with the healthcare systems and value-based care focus of the manuscript, which we believe is an important aspect of translational implementation in comprehensive cancer centers. We hope this addresses the concern while reaffirming that the manuscript’s current content is consistent with its intended scope.

Round 2

Reviewer 1 Report

Comments and Suggestions for Authors

The paper has greatly improved deserving publication.

Reviewer 2 Report

Comments and Suggestions for Authors

My opinion about the manuscript has not changed. I leave the decision to the academic editor to accept/reject the manuscript. In my opinion, this is not research.